# Association between psychological vulnerability and glaucoma progression: Protocol for a multicenter prospective cohort study in South Korea

**Sung Uk Baek[1,2]☉, Jin-Soo Kim[3]☉, Dai Woo Kim[4], Ahnul Ha[5,6]‡, Young Kook Kim**👁[7,8,9]‡*

**1** Department of Ophthalmology, Hallym University College of Medicine, Anyang, Korea, **2** Department of Ophthalmology, Hallym University Sacred Heart Hospital, Anyang, Korea, **3** Department of Ophthalmology, Chungnam National University Sejong Hospital, Sejong, Korea, **4** Department of Ophthalmology, School of Medicine, Kyungpook National University, Daegu, South Korea, **5** Department of Ophthalmology, Jeju National University Hospital, Jeju-si, Korea, **6** Department of Ophthalmology, Jeju National University School of Medicine, Jeju-si, Korea, **7** Department of Ophthalmology, Seoul National University Hospital, Seoul, Korea, **8** Department of Ophthalmology, Seoul National University College of Medicine, Seoul, Korea, **9** EyeLight Data Science Laboratory, Seoul, Korea

☉ These authors contributed equally to this work.
‡ AH and YKK also contributed equally to this work as co-corresponding authors.
* md092@naver.com

**Data Availability Statement:** Deidentified research data will be made publicly available when the study is completed and published.

## Abstract

### Introduction

Mental health is a significant concern for patients diagnosed with glaucoma, as visual impairment can have a profound impact on psychological well-being. Conversely, psychological vulnerability in glaucoma patients can negatively influence treatment adherence to ocular hypotensive therapy, thereby potentially exacerbating disease progression in a vicious cycle. The study protocol proposed herein aims to explore the impact of psychological states such as anxiety, depression, and stress on both medication adherence and progression of glaucoma.

### Materials and methods

This study is to be a prospective multicenter study conducted at four tertiary medical centers. Patients recently diagnosed with glaucoma and not yet treated will be enrolled. Anxiety, depression, and stress scales will be administered at baseline, one year, and two years, along with glaucomatous assessments to be performed every six months. Validated questionnaires (Generalized Anxiety Disorder Assessment [GAD-7], Patient Health Questionnaire [PHQ-9], and Perceived Stress Scale-10 [PSS-10]) will assess anxiety, depression, and stress, respectively. The primary objective is to correlate post-diagnosis psychological status with medication adherence and disease progression. The effects of pre- and post-diagnosis changes in anxiety, depression, and stress on disease progression will be analyzed. Kaplan-Meier survival analysis and logistic regression will be performed to identify

**Funding:** The authors received no specific funding for this work.

**Competing interests:** The authors have declared that no competing interests exist.

clinical characteristics associated with increased risk of developing anxiety, depression, and stress in glaucoma patients.

## Introduction

Glaucoma is a vision-threatening disorder that potentially has a significant impact on patients' quality of life [1, 2]. Visual field (VF) loss affects visual functions such as stereopsis and contrast sensitivity, and glaucoma patients face higher risks of hospitalization, high treatment costs, and accidents [3–5]. Consequently, newly-diagnosed glaucoma patients often experience anxiety about their future.

Studies show that visual-impairment-related stress predicts depressive symptoms, regardless of objective visual acuity measurement [6, 7]. Anxiety and depression are frequently associated with glaucoma, with prevalence rates ranging from 6 to 25% and 13 to 30% in primary open-angle glaucoma (POAG) patients, respectively [8]. Glaucoma remains a significant predictor of depression, even after adjusting for demographic factors and comorbidities [8, 9]. Depression is linked to longer follow-up periods and more severe disease progression, while faster VF deterioration is associated with depressive symptoms [10, 11].

The role of anxiety and depression in glaucoma is complex. They can be consequences of visual impairment and glaucoma diagnosis, but, in a kind of vicious cycle, may also contribute to disease progression. Specifically, depression reduces medication adherence, which is known to accelerate VF progression [12, 13]. Stress, meanwhile, has been found to increase intraocular pressure (IOP) in non-human primates, potentially influencing the relationship between anxiety and glaucoma [14].

Given the multifactorial nature of emotional disorders and glaucoma, understanding their connection is challenging. A longitudinal study using a multicenter cohort database is needed to explore how psychological states change and impact both medication adherence and disease progression. Therefore, the study protocol proposed herein was devised with the aim of evaluating the influence of anxiety, depression, and stress on adherence to ocular hypotensive therapy and glaucoma progression.

## Methods

### Ethics and declaration

This study was approved by the relevant Institutional Review Boards and informed consent will be obtained. The study protocol has been registered in the Open Science Framework (OSF) under registration number DOI 10.17605/OSF.IO/DA2F4.

### Study design

The proposed prospective multicenter cohort study will have a 2-year follow-up period. Patient recruitment is planned to commence in November 2023 after study-protocol registration. Analysis is expected to be completed by the end of 2024. Anxiety, depression, and stress scales along with glaucomatous assessments will be collected at each follow-up visit. The study will be conducted simultaneously at four nationwide medical institutions in South Korea: Seoul National University Hospital, Hallym University Sacred Heart Hospital, Chungnam National University Sejong Hospital, and Jeju National University Hospital. These institutions are geographically widely distributed in different cities and provinces. All of the participating

centers have appropriate facilities for glaucoma diagnosis and treatment as well as emotional disorder assessment. According to prior training, clinicians at each institution will try to use a standardized method of delivering the diagnosis of glaucoma. All participants will be treated by a single physician throughout the follow-up period at each institution.

## Subjects

The study flowchart, including the schedules of ophthalmologic examinations and questionnaires, is presented in Fig 1. Informed consent will be obtained from all participants. Only one eye per patient will be included, and if both eyes meet the eligibility criteria, one eye will be randomly included.

  The inclusion and exclusion criteria are outlined in Table 1. Newly diagnosed glaucoma patients who are treatment-naive will be included in the study. Patients will have to meet specific criteria for open-angle glaucoma, including glaucomatous optic disc changes, retinal nerve fiber layer (RNFL) defects, glaucomatous visual field (VF) defects, and open angle as confirmed by gonioscopic examination. Glaucomatous eyes were defined as those showing glaucomatous optic disc neuropathy (e.g., notching, neuroretinal rim thinning, and/or RNFL defects) and corresponding glaucomatous VF defects, as confirmed by at least two consecutive VF examinations. Glaucomatous VF defects were defined as a cluster of $\geq 3$ points with $P < 0.05$ on the pattern deviation map in at least one hemifield, including $\geq 1$ point with $P < 0.01$; a pattern standard deviation of $P < 0.05$; or glaucoma hemifield test result outside the normal limits. VF defects will be confirmed based on reliable tests (fixation loss rate $\leq 20\%$, false positive and false negative error rates $\leq 25\%$). Patients with a history of

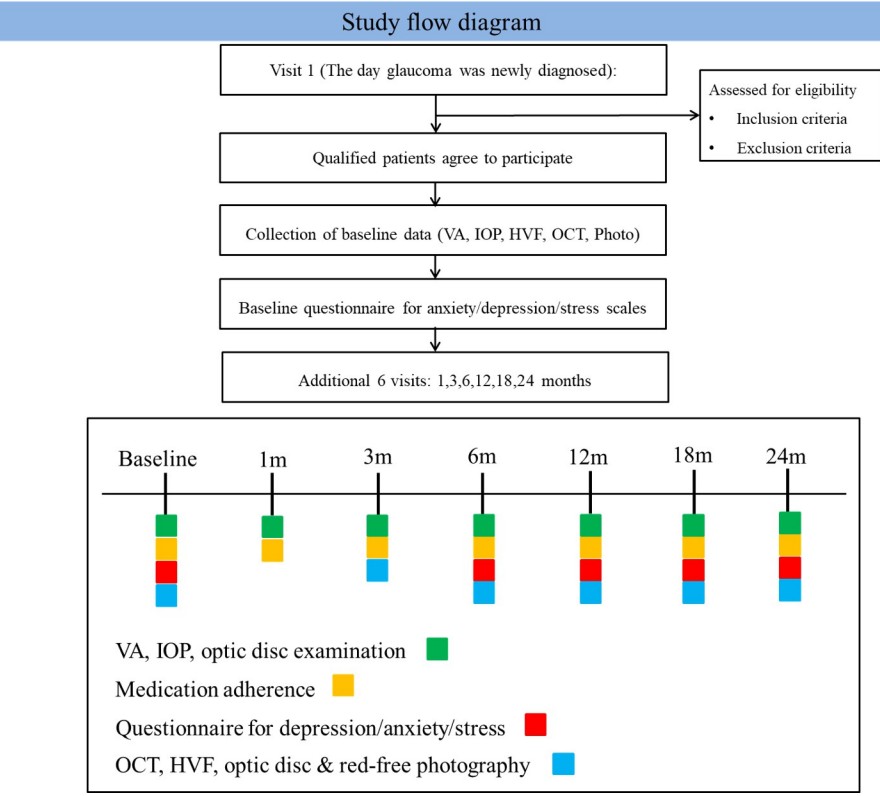

**Fig 1. Flowchart of enrollment.**

intraocular surgery (other than uncomplicated cataract extraction more than 6 months previously) or retinal pathology will be excluded. Participants with a history of neurodegenerative disease or cerebrovascular accidents that may affect visual acuity or VF examination results also will be excluded. Patients with diabetes mellitus or systemic hypertension will be included unless already diagnosed with diabetic or hypertensive retinopathy (**Table 1**). No subjects will receive any grants, bonuses or incentives for participating in the study.

## Ophthalmologic assessments

All enrolled patients will undergo a comprehensive ophthalmological examination, including assessment of best-corrected visual acuity (BCVA), slit lamp biomicroscopy, intraocular pressure (IOP) measurement by Goldmann applanation tonometry (Haag-Streit), anterior chamber angle measurement by gonioscopy, stereoscopic examination of the optic disc, red-free fundus photography (TRC-50IX; Topcon Corporation), central corneal thickness measurement with ultrasound pachymetry, and axial length measurement by optical biometer. RNFL thickness will be measured by Cirrus optical coherence tomography (OCT) (software version 9,5; Carl Zeiss Meditec, Dublin, CA, USA), and Humphrey visual field (HVF) (Humphrey Field Analyzer II; 24–2 Swedish Interactive Threshold Algorithm; Carl Zeiss Meditec). testing will be performed using Swedish Interactive Threshold Algorithm standard 24–2 perimetry. In the initial evaluation, consecutive VF tests will be performed on a single day or the other separate day to secure two or more reliable baseline VF tests. The same optic disc, red-free fundus photography, OCT and HVF models will be used across all of the research institutions. Visual acuity testing, IOP measurement, and optic disc examination will be conducted at each visit, while OCT, HVF, optic disc, and red-free photography will be evaluated at 3 months and every 6 months (**Fig 1**). The presence of optic disc hemorrhage will be assessed by direct observation using a slit lamp or photography.

## Anxiety/depression/stress scales

Anxiety, depression, and stress levels will be assessed on standardized scales: the Generalized Anxiety Disorder Assessment (GAD-7), the Patient Health Questionnaire (PHQ-9), and the Perceived Stress Scale-10 (PSS-10), respectively. The GAD-7, PHQ-9, and PSS-10 scales are

**Table 1. Inclusion and exclusion criteria.**

| **Inclusion criteria** |
| --- |
| • Aged >18 years old. |
| • Voluntary participation in the study and providing informed consent.<br>• Newly diagnosed glaucoma patients who have not received any treatment.<br>• Open-angle glaucoma characterized by open angles, glaucomatous optic nerve damage, and confirmed visual field damage. |
| **Exclusion criteria** |
| • Other ophthalmic diseases that may impact visual function.<br>• High myopia (defined as spherical equivalent > −6.0 diopters).<br>• Previous intraocular surgery (other than uncomplicated cataract extraction more than 6 months previously)<br>• Hospitalization or severe illness within the past 6 months or during the study period.<br>• Significant systemic diseases such as degenerative central nervous system disease, brain tumor, cerebral hemorrhage, or cerebral ischemia.<br>• Expected difficulty in completing the questionnaire due to cognitive impairment (e.g., old age, dementia).<br>• Severe mental illness such as schizophrenia or severe dementia.<br>• History of bipolar disorder, depressive or anxiety disorders prior to glaucoma diagnosis.<br>• Currently taking antipsychotic, antidepressant or antianxiety medications.<br>• Pregnancy or lactation at enrollment. |
| • Experience of a personal trauma such as death of a loved one, job loss or divorce within the previous 6 months. |

short and understandable screening instruments that are useful for detecting and assessing severity of symptoms in clinical and research settings. In addition, these scales have been formally validated against diagnostic clinical interviews to establish the sensitivity, specificity and positive and negative predictive values of cut-off scores, and are used widely in clinical research [15–17]. To that end, self-rating questionnaires will be administered at baseline and at every 6-month follow-up visit (**Fig 1**). Validated Korean versions of the questionnaires will be used [18–20]. Participants will be able to complete the questionnaires at the clinic or verbally if necessary. Consultation with a psychiatrist will facilitate interpretation of the survey results.

**1) Anxiety scale.** The GAD-7 consists of seven items that measure anxiety symptoms experienced in the preceding two weeks. Each item is scored from 0 to 3 based on symptom frequency, ranging from "none" to "almost every day." The total score ranges from 0 to 21, with higher scores indicating more frequent symptoms. The GAD-7 is reliable and valid for identification of anxiety disorders such as generalized anxiety disorder (GAD) [21, 22].

**2) Depression scale.** The PHQ-9 is derived from the nine symptoms of depression outlined in the *Diagnostic and Statistical Manual of Mental Disorders*, 4th Edition (*DSM-IV*) criteria. This scale is a concise and reliable tool for screening and assessment of depression severity. The scoring system is the same as that of the GAD-7, with a score over 15 indicating severe depression [15, 23].

**3) Stress scale.** The PSS-10 is a validated self-administered questionnaire widely used for assessment of perceived stress [24]. It assesses moods and thoughts experienced in the preceding month, focusing on general stressors and coping abilities. Participants rate the frequency of encountering such situations. The scoring system is the same as those of the GAD-7 and PHQ-9, with higher scores indicating higher perceived stress. A score between 0 and 14 is considered to reflect mild stress, while scores above 15 indicate severe stress [25].

## Psychiatric management

If one or more of a patient's GAD-7, PHQ-9, and PSS scores indicate a severe stage, additional visits to a psychiatry specialist will be scheduled. Psychiatrists will provide further evaluation and treatment recommendations. Treatment for depression or anxiety is based on the patient's preference and is not mandatory. Information regarding antidepressant or antianxiety treatments will be recorded in Case Report Forms (CRFs).

## Medication adherence

After glaucoma diagnosis, drug treatment will begin. Topical antiglaucoma medications will be selected and dosages will be determined based on glaucoma severity, baseline IOP, and patient factors. In this study, a target IOP showing at least a 20–30% reduction from the initial pressure will be set, and clinicians at each institution will use it as a guideline for lowering of IOP [26]. Clinicians may adjust medications if IOP control is insufficient or if glaucoma progression occurs during follow-up. The target IOP will be reassessed periodically and lowered if progression, optic nerve hemorrhage, or increase in risk factors occurs.

To assess adherence, we will measure the total number of weekly eye-drop administrations and instances of missed doses over the preceding 7 days [27]. The adherence rate will be

calculated using the following formula:

Medication adherence rate (%)

$$= \frac{([\text{Daily total N of PD}] \times 7 \text{ days} - \text{N of PD missed during preceding week})}{\text{Weekly total N of PD}} \times 100$$

*N = number; PD = Prescription doses.*

As indicated in the formula, the total number of weekly topical administrations will be obtained by multiplying the patient's self-reported daily number of prescribed eye-drop doses (PD) by seven. Patient self-reporting will be used to assess the number of missed doses in the preceding seven days. This number will be subtracted from the total PD for the week. The resulting value will be divided by the weekly total PD, and then multiplied by 100 to determine the medication adherence rate (%). Patients with an adherence rate $\geq 80\%$ will be considered to be *adherent*, while those with an adherence rate $<80\%$ will be classified as *non-adherent* [28].

## End points

End points requiring discontinuation in the study included: treated IOP >21 mmHg despite maximal tolerated medical therapy on two successive occasions (safety end point); glaucoma surgery or glaucoma laser treatment (laser iridotomy, laser iridoplasty, laser trabeculoplasty) (therapeutic end point); pregnancy or lactation during follow-up; and missing two or more consecutive questionnaire responses. Data up to this point were included in the analysis, but discontinued patients were no longer followed as participants.

## Analysis of psychiatric scales and glaucoma progression

**Primary objectives.** The main objective of this study is to investigate the relationship between an individual's psychiatric state (as measured on anxiety/depression/stress scales) and progression of glaucoma. We also aim to analyze how anxiety, depression and stress levels affect medication adherence during glaucoma treatment.

**Secondary objectives.** The secondary objectives are (1) to evaluate speed of glaucomatous progression according to psychological scales, (2) to identify psychological factors for medication non-adherence, and (3) to identify ophthalmic factors for psychological deterioration.

The detailed study-outcomes to be investigated can be summarized as follows:

- The association between the initial anxiety/depression/stress scales and the severity of glaucoma at the time of diagnosis.

- The correlation between the initial anxiety/depression/stress scales and the glaucoma progression rate or medication adherence.

- The longitudinal change pattern of psychiatric scale between progression and non-progression group.

- Changes in medication adherence and psychological scale according to follow-up

- Risk factors for psychological deterioration and medication non-adherence in glaucoma patients.

## Glaucoma progression assessment

During the 2-year follow-up period, glaucoma progression and rate will be evaluated. Structural assessments (RNFL thickness, RNFL defect, disc hemorrhage, optic disc rim change) and functional assessments (VA, VF) will be performed, and corresponding changes will be recorded.

Glaucoma progression will be determined by observing structural and functional changes. The markers to be included are rim notching or thinning, cup contour alteration, optic disc vasculature change, and increased cup-to-disc ratio. Progressive pathologic changes in the optic disc will be evaluated by comparison of serial color disc photographs. RNFL progression will be identified by defect depth/width and new defects. Optic disc features are comprehensively evaluated by checking the related RNFL profile (red-free photo and OCT). VF progression will be evaluated using "event-based" and "trend-based" analyses. The event-based analysis will employ the Humphrey field analyzer to detect significant pattern deviation decrease at multiple test points on consecutive tests. The trend-based analysis will calculate progression rate based on mean deviation (MD) change over time.

During the research period, data will be accumulated without any evaluation of progress. The final trial dataset will be blinded to any identifying participant information and uploaded to an open-access data repository. The analysis probably will be completed after the 2-year study period. At the end of the study, each participant's clinical data will be assessed for progress.

## Sample size calculation

The sample size was based on the outcome of the Early Manifest Glaucoma Trial (EMGT): [29] the glaucoma progression rates at 24 months were 20% in the untreated group and 10% in the treated group, and IOP reduction was 25% in the treated group. In this study, baseline-target IOP was set at 20 to 30% IOP reduction, depending on baseline IOP. Based on 90% power (1−β) at a 2-sided error α = 0.05 to detect the difference between 20 and 10% in incident progression over a 24-month follow-up, 125 patients will be needed in each arm (250 total). Allowing for 20% attrition over the study period, 300 patients will need to be recruited for the study.

## Masked and blinded evaluation

The masked and blinded evaluation for minimization of bias can be summarized as follows:

- Masked technicians who obtain the IOP measurements are not masked to the patient's participation in the study.

- Only the study designers/allocators know the patients' initial psychological scores, and the physicians are blinded to the scores.

- Allocators who decide on the need for psychiatric intervention in patients are independent and blinded to the stage of glaucoma.

- Investigators blindly perform the statistical analyses.

## Statistical analysis plan

A descriptive analysis will be performed on potential risk factors for glaucoma progression. Continuous variables will be presented with mean/SD or median/interquartile values based on data distributions. Categorical variables will be reported in frequency tables (n, %).

Parametric/nonparametric tests will be conducted based on data normality (Kolmogorov-Smirnov test).

All covariates will be collected at baseline. Patients will be grouped into five categories based on age: <40, 40–49, 50–59, 60–69, and ≥70 years. Sex will be categorized as male or female. Health insurance will be categorized based on the Korean National Health Insurance System as employee, self-employed individual, or medical aid beneficiary. Residence regions will be grouped into metropolitan area (Seoul, Busan, Daegu, Incheon, Daejeon, Gwangju, Ulsan) or local province (Gangwon, Gyeonggi, Chungcheongbuk, Chungcheongnam, Sejong, Gyeongsangbuk, Gyeongsangnam, Jeollabuk, Jeollanam, Jeju).

Anxiety, depression, and stress scale severity will be categorized into Mild (GAD-7 < 11) vs. Severe anxiety (GAD-7 ≥11), Mild (PHQ-9 < 15) vs. Severe depression (PHQ-9 ≥15), and Mild (PSS-10 < 15) vs. Severe stress (PSS-10 ≥ 15). 'Adherent patient' (adherence ≥ 80%) and 'Non-adherent patient' (adherence < 80%) groups will be compared by t-test/chi-square testing.

Survival analysis by the Kaplan-Meier method will assess glaucoma progression probability. Logistic regression will identify clinical characteristics associated with anxiety/depression/stress and glaucoma progression. Multivariable models will adjust for confounding factors. Variables with $P < 0.1$ (univariate model) will be included (significance level: 2-sided $P < 0.05$). SPSS software (ver. 27.0; SPSS Inc., Chicago, IL, USA) will be used.

## Discussion

This prospective and longitudinal study will identify changes in patients' psychological state and objectively demonstrate the potential impact of these factors on disease progression.

### Strengths

Previous studies have focused primarily on cross-sectional analyses of the association between anxiety, depression, and stress scales and glaucoma [8, 10, 30, 31]. By contrast, the study protocol proposed herein will include a longitudinal approach to prospectively investigate the effects of anxiety, depression, and stress on glaucoma progression. Further, it will allow for causal relationship analysis, by which the influence of clinical factors such as glaucoma severity, progression rate, and medication adherence on patients' psychological state can be explored. Overall, by establishing a cohort group consisting of newly diagnosed glaucoma patients, this study will be able to longitudinally analyze changes in their mental health measures during follow-up.

Drawing a parallel to Kübler-Ross's model describing psychological changes in dying patients [32], this study will examine the psychological changes experienced by glaucoma patients during initial diagnosis, treatment, and follow-up. Additionally, as a multicenter study, it will enhance the generalizability of the findings by including a diverse patient group and enabling the analysis of factors related to patients' disease progression and various circumstances.

### Limitations

First, as a multicenter study, variations in patient treatment, medication usage, and doctor-patient relationships may exist due to different clinician styles. Second, the questionnaire itself may not fully capture the complexity of patients' psychological state and could lead to more positive responses. Relying on self-reported anxiety, depression and stress questionnaires might result in underrepresentation of the true prevalence of these symptoms. Moreover, the study will include only participants who voluntarily agree to participate in the questionnaire,

thus introducing the potential for selection bias and variations in response rates (i.e., patients with severe depression or anxiety may naturally be less willing to participate). Third, using patient-reported adherence might not be a reliable way to assess adherence, as most patients have a tendency to overestimate their adherence to medication. Fourth, socioeconomic and environmental factors can influence patients' psychological state, treatment adherence, and access to healthcare, which fact might introduce confounding variables that limit the establishment of a causal relationship between psychological distress and glaucoma progression. Fifth and finally, the study, as it will be conducted at tertiary medical centers, may include a population that exhibits lower levels of anxiety or depression, due to prior exposure to glaucoma diagnosis and related information from primary and secondary healthcare facilities.

## Supporting information

**S1 Checklist. STROBE statement—checklist of items that should be included in reports of observational studies.**
(PDF)

## Author Contributions

**Conceptualization:** Sung Uk Baek, Jin-Soo Kim, Ahnul Ha, Young Kook Kim.

**Data curation:** Sung Uk Baek, Ahnul Ha, Young Kook Kim.

**Formal analysis:** Sung Uk Baek, Jin-Soo Kim, Ahnul Ha, Young Kook Kim.

**Investigation:** Sung Uk Baek, Young Kook Kim.

**Methodology:** Dai Woo Kim, Ahnul Ha, Young Kook Kim.

**Project administration:** Sung Uk Baek, Young Kook Kim.

**Resources:** Sung Uk Baek, Jin-Soo Kim, Dai Woo Kim, Ahnul Ha, Young Kook Kim.

**Software:** Sung Uk Baek, Jin-Soo Kim, Ahnul Ha, Young Kook Kim.

**Supervision:** Sung Uk Baek, Jin-Soo Kim, Dai Woo Kim, Ahnul Ha.

**Validation:** Sung Uk Baek, Jin-Soo Kim, Dai Woo Kim, Ahnul Ha, Young Kook Kim.

**Visualization:** Sung Uk Baek, Jin-Soo Kim, Ahnul Ha, Young Kook Kim.

**Writing – original draft:** Sung Uk Baek, Young Kook Kim.

**Writing – review & editing:** Sung Uk Baek, Ahnul Ha, Young Kook Kim.

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
