## [Decision Letter · Decision Letter 0]

25 Jul 2023

PONE-D-23-18243Association Between Psychological Vulnerability and Glaucoma Progression: Protocol for a Multicenter Prospective Cohort Study in South KoreaPLOS ONE

Dear Dr. Kim,

Thank you for submitting your manuscript to PLOS ONE. After careful consideration, we feel that it has merit but does not fully meet PLOS ONE’s publication criteria as it currently stands. Therefore, we invite you to submit a revised version of the manuscript that addresses the points raised during the review process.

Four reviews gave their comments.  I agree that with minor revisions, this manuscript is going to be acceptable for publication.The revisions needed are:a more in depth explanation of inclusion and exclusion criteria, specially in regards to newly diagnosed glaucoma patients and cataract surgery. It is important to define the demographic characteristics to be analyzed, as these need to be standardized. 

Criteria for Visual field analysis needs to be more explicit. Reliability cut-points, definition of damage, and ways to standarice between different sites of the study. 

All 4 reviewers asked for a clarification of the sedition to only take one eye ( either best or worst). Is there an explanation as to whether one  eye may impact the patients differently?

A more thorough explanation on the statistics of the sample size are needed. 

 Lastly, one of the revierwers noted that :"The authors provide two OSF registration numbers in this manuscript. Please confirm which one is correct. Is it DOI 10.17605/OSF.IO/AWTEC (mentioned in the manuscript information) or DOI 10.17605/OSF.IO/DA2F4 (mentioned in the main text)? " 

This point needs to be clarified, as any inconsistencies with the registration, will deem the manuscript not apt for publication.

We look forward to receiving your revised manuscript.

Kind regards,

Alvaro Jose Mejia-Vergara, MD

Academic Editor

PLOS ONE

Journal Requirements:

Additional Editor Comments:

Three reviewers find only minor revisions needed. I do agree with these reviews. The last reviews states mayor revisions needed, but can be achieved easily with minor changes.

Reviewers' comments:

Reviewer's Responses to Questions

**Comments to the Author**

1. Does the manuscript provide a valid rationale for the proposed study, with clearly identified and justified research questions?

Reviewer #1: Yes

Reviewer #2: Yes

Reviewer #3: Yes

Reviewer #4: Yes

2. Is the protocol technically sound and planned in a manner that will lead to a meaningful outcome and allow testing the stated hypotheses?

Reviewer #1: Yes

Reviewer #2: Partly

Reviewer #3: Yes

Reviewer #4: Partly

3. Is the methodology feasible and described in sufficient detail to allow the work to be replicable?

Reviewer #1: Yes

Reviewer #2: Yes

Reviewer #3: Yes

Reviewer #4: No

4. Have the authors described where all data underlying the findings will be made available when the study is complete?

Reviewer #1: Yes

Reviewer #2: Yes

Reviewer #3: Yes

Reviewer #4: No

5. Is the manuscript presented in an intelligible fashion and written in standard English?

Reviewer #1: Yes

Reviewer #2: Yes

Reviewer #3: Yes

Reviewer #4: Yes

6. Review Comments to the Author

You may also provide optional suggestions and comments to authors that they might find helpful in planning their study.

Reviewer #1: “Association Between Psychological Vulnerability and Glaucoma Progression: Protocol for a Multicenter Prospective Cohort Study in South Korea”

The study protocol is well structured and written.

Query 1: Minor language revision (e.g. “words counts” instead of word count, “cerebral infarction” instead of cerebral ischemia, Table 1 (no period at the end of each phrase…). Please revise.

Query 2: Table 1. Exclusion criteria, uncontrolled IOP > 21 despite medication. Newly diagnosed glaucoma is the main inclusion criteria highlighted by the authors, therefore this exclusion criteria seems to be redundant. Please explain.

Query 3: Patients with uncomplicated cataract surgery can be included in the study? There seems to be a contradiction, a following paragraph states that a history of cataract during the previous 6 months is an exclusion criteria. Please, explain.

Query 4: Ophthalmologic assessment. The authors should provide the model of each device used in the study. All the participant centers should use the same model of each device.

Query 5: “Glaucoma progression will be determined by observing structural and functional changes. The markers to be included are rim notching or thinning, cup contour alteration,voptic disc vasculature change, and increased cup-to-disc ratio. RNFL progression will be identified by defect depth/width and new defects”. How will these changes be reported by each investigator?

Query 6. The authors mentioned other cross-sectional and retrospective studies about the subject. Those studies included a very wide number of cases (6.760 to more than 4 millions…). The authors plan to include 300 patients, but I think they didn´t provide the statistical method to calculate the sample size.

Reviewer #2: Review: Association Between Psychological Vulnerability and Glaucoma Progression: Protocol for a Multicenter Prospective Cohort Study in South Korea

The authors aim to study a very important facet of glaucoma which is often overlooked: the psychologic impact of this disease on our patients. Importantly, they also aim to examine the role of psychologic diagnoses in glaucoma worsening via its role in medication noncompliance. I commend the authors for identifying this area of much needed research and designing a study to gather this information.

A large part of psychologic vulnerability and glaucoma could be impacted by the physician patient relationship. Is there standard way patients are told about their disease? Is the education provided standardized for each patient enrolled? Some physicians resort to scare tactics during diagnosis which causes increased anxiety in patients and this could influence results and the patient’s reaction to their diagnosis.

Methods: subjects

What is your definition of a reliable visual field test, this should be defined in the protocol.

Will background demographics for each patient be collected? This study will need to control for age, sex, socioeconomic status, education level, transportation, employment status, social support network etc as these may play an important role in psychologic response to diagnosis. I assume this will be collected but it is not explicitly stated.

Is there a standard for treatment? How can you control for inadequate treatment causing progression versus medication noncompliance? Will patients all be started on similar therapies? Is there a standard percentage of IOP lowering that will be targeted as a guide for initial treatment or is this left to the discretion of the treating provider?

Medication Adherence:

Using patient reported adherence is a very poor way to assess adherence. Most patients overestimate their adherence to medication. Pharmacy refill data or devices which track adherence are a much more reliable measure of medication adherence in this technological climate. How will patients keep track of this? Many patients do not recall they have missed doses?

Analysis of Glaucoma Progression

In newly diagnosed glaucoma patients, tracking patients over 2 years may not provide enough data (with 4 visual fields? i.e. one every 6 months) to determine progression. It will give an idea but often there is a learning curve in field testing.

Reviewer #3: The idea of this protocol is excellent and nowadays very popular. Also the paper contains interesting findings, however, there are following issues to be addressed:

1. Only patients with open-angle glaucoma (primary open-angle glaucoma [POAG] and normaltension glaucoma [NTG]) will be included in the study? And if so, why?

2. Why only one eye per patient will be included, or if both eyes are eligible, the eye with the more severe mean deviation (MD) value from the visual field test?

3. Whether experience of a personal trauma such as death of a loved one, job loss or recent divorce will be taken into account as an exclusion criterion?

4. Whether psychiatry specialist will be included in the study and for the interpretation of the results?

5. Why were the scales: the Generalized Anxiety Disorder Assessment (GAD-7), the Patient Health Questionnaire (PHQ-9) chosen to assess anxiety and depression?

6. Out of 25 references, only three are 5 years old.

Reviewer #4: This article reports on the study protocol of a prospective multicenter cohort study designed to investigate the impact of psychological states such as anxiety, depression, and stress on both medication adherence and glaucoma progression. This cohort study, if successfully conducted and completed, will contribute to the understanding of changes in the psychological state of glaucoma patients during the course of treatment and provide valuable clinical information for the psychiatric management of patients with glaucoma. However, the study protocol reported in this article is incomplete and needs to be revised (see comments below).

Comments:

1. Ethics Statement:

The authors provide two OSF registration numbers in this manuscript. Please confirm which one is correct. Is it DOI 10.17605/OSF.IO/AWTEC (mentioned in the manuscript information) or DOI 10.17605/OSF.IO/DA2F4 (mentioned in the main text)?

2. Subjects

① “…if both eyes are eligible, the eye with the more severe mean deviation (MD) value from the visual field test will be selected.”

I would like to know whether it is the worse eye or the better eye that has a greater impact on the patient's psychological state.

If a participant with glaucoma reaches legal blindness in one eye and still has better vision in the other eye, say 20/30, he/she may be in two states of mind, fearing deterioration in the better-sighted eye or, living unaffected because he/she still has one eye with adequate vision. I wonder if the worse eye or the better eye could have a greater impact on the patient's psychological states.

If the conclusions in this aspect are controversial, I would suggest that if both eyes meet the eligibility criteria, one eye be randomly included.

② “VF defects will be confirmed based on specific criteria, and reliable tests will be conducted.”

Please add the definition of glaucomatous VF damage and the criteria of reliable tests.

3. Inclusion and exclusion criteria

① “Patients with a history of ocular surgery (except uncomplicated cataract surgery) …”

“Table 1:

Exclusion criteria:

…

• History of cataract, glaucoma, or retina surgery within the past 6 months

or during the study period …”

Please maintain consistency of statements.

② “Patients will have to meet specific criteria for open-angle glaucoma (primary open-angle glaucoma [POAG] or normal tension glaucoma [NTG]), including…”

“Table 2:

…

• Open-angle glaucoma characterized by open angles, glaucomatous optic nerve damage, and confirmed visual field damage”

Please maintain consistency of statements. Is secondary open-angle glaucoma excluded?

③ “Exclusion criteria:

• Uncontrolled intraocular pressure (> 21 mmHg) despite medication”

Since the participants are all newly diagnosed with glaucoma, it will take at least some time to see if the IOP can be controlled with medication alone. What is the approximate length of this period? Will the baseline data be collected before or after this period?

④ Is undergoing glaucoma surgery an indicator of glaucoma progression, or the endpoint of this study?

⑤ Study exclusion criteria include pregnancy and lactation. During the enrollment process, will participants be informed of the need to use contraception or breastfeed during the 2-year study?

⑥ If a participant has experienced a vital event within the 6 months or during the study period, e.g., birth, death, marriage/civil partnership, divorce, adoption, would he or she be excluded? These life events are known to affect a person's social behavior and mental state. How will the researcher investigator deal with the impact of these events on the study results?

⑦ I was curious that high myopia may also lead to some structural changes in the fundus similar to POAG, especially in the optic disc area. Based on the long follow-up time proposed in this study, I suggest that diopter or axial length restrictions be added to the exclusion criteria. If not, please give a reason that can be identified.

⑧ Will a participant be treated by the same physician throughout the follow-up period?

⑨ Please indicate if the participant will receive any grants, bonuses or incentives.

4. Trial design

① Please state whether this prospective study involves blinding:

a. For example, only the study designers/allocators knew the patients' initial mental scores and the physicians were blinded to the scores. This may have allowed physicians to avoid giving more aggressive treatment to patients with lower scores.

b. Are the allocators who decide on the need for psychiatric intervention in patients independent and blinded to the stage of glaucoma?

c. Are the investigator(s) performing the statistical analyses blinded?

d. If this study is not blinded, what means will the investigator use to reduce the associated bias?

② Given the long follow-up proposed in this study, how will missing responses be handled?

③ Does this study include a control population?

5. Anxiety/Depression/Stress Scales

I suggest that the questionnaires also be administered at the 6-month and 18-month follow-up visits. The Kübler-Ross model has five successive stages. If it is true, as the authors suggest, that the mental status of participants after glaucoma diagnosis is similar to the Kübler-Ross model, then administering the questionnaire only at baseline, 1-year, and 2-year follow-up visits may be insufficient.

6. Analysis of Psychiatric Scales and Glaucoma Progression

① Please state the primary, secondary and exploratory objectives (if any) and safety objectives of this study.

② The first and second points of the study results appear to be repetitive.

③ The described study outcomes focus on the correlation between the initial scale and glaucoma. There is a lack of study outcomes on the dynamic changes during the course of the study.

7. Glaucoma Progression Assessment

“Glaucoma progression will be determined by observing structural and functional changes. The markers to be included are rim notching or thinning, cup contour alteration, optic disc vasculature change, and increased cup-to-disc ratio.”

① Are changes in the cup-to-disc ratio based on the subjective judgment of the clinician or the objective analysis of the testing instrument?

② Please subpoint the conditions that are considered glaucoma progression, as this appears to be the endpoint of this study.

8. Statistical Analysis Plan and Handling Bias

① “This multicenter study estimates that about 300 participants would be needed, assuming the accuracy and reliability of the questionnaire to be 80% with the target number of sample size.”

Please explain the sample size calculation. What is the assumption? Is the test one- or two-sided? How much statistical power does the sample size provide? And please provide references for these assumed data.

② Jeju-do has a much smaller population than the Seoul area and Chungcheongnam-do. And two of the four hospitals are located in the Seoul area. What is the expected proportion of participants to be enrolled in each of the four centers? And how can uneven enrollment be avoided?

③ Please briefly describe how the researcher will reduce the biases.

9. Ethics and Declaration Please place this section before or merge it with the Study Design section.

7. PLOS authors have the option to publish the peer review history of their article (what does this mean?). If published, this will include your full peer review and any attached files.

Reviewer #1: No

Reviewer #2: No

Reviewer #3: No

Reviewer #4: No

---

## [Author Response · Author response to Decision Letter 0]

7 Sep 2023

Submission ID: PONE-D-23-18243

Manuscript title: Association Between Psychological Vulnerability and Glaucoma Progression: Protocol for a Multicenter Prospective Cohort Study in South Korea

Response to the Reviewers: Point-by-point response to the comments

Editor’s comments:

Four reviews gave their comments. I agree that with minor revisions, this manuscript is going to be acceptable for publication. 

The revisions needed are: a more in depth explanation of inclusion and exclusion criteria, specially in regards to newly diagnosed glaucoma patients and cataract surgery. It is important to define the demographic characteristics to be analyzed, as these need to be standardized. 

Criteria for Visual field analysis needs to be more explicit. Reliability cut-points, definition of damage, and ways to standarize between different sites of the study. 

All 4 reviewers asked for a clarification of the sedition to only take one eye ( either best or worst). Is there an explanation as to whether one eye may impact the patients differently?

A more thorough explanation on the statistics of the sample size are needed. 

Lastly, one of the reviewer noted that :"The authors provide two OSF registration numbers in this manuscript. Please confirm which one is correct. Is it DOI 10.17605/OSF.IO/AWTEC (mentioned in the manuscript information) or DOI 10.17605/OSF.IO/DA2F4 (mentioned in the main text)? " 

This point needs to be clarified, as any inconsistencies with the registration, will deem the manuscript not apt for publication.

:The authors appreciate the Editor for giving us the opportunity to revise this manuscript. Implementation of the suggested edits has improved its overall presentation and clarity. During the revision process, we modified the inclusion and exclusion criteria. In addition, we added contents for reliable visual field tests and a definition of glaucomatous scotoma. In the case of binocular glaucoma, we have amended the inclusion criteria to randomly select only one eye. Next, we added a statistical explanation to calculate the sample size, and provided references. Lastly, we have made the DOI consistent throughout the manuscript. 

In this revision, we have addressed all these comments and highlighted all of the revisions in blue color. We hope the revised manuscript has now met the publication standard of your journal. Our point-to-point responses to the queries raised by the editor and reviewers are listed below. 

Reviewer #1

# General comments:

The study protocol is well structured and written.

: We greatly appreciate the reviewer's corrections and requests. The authors fully agree with all of the requests and believe that this manuscript has been revised to make it more solid and well-organized due to your correction. Thank you very much again.

# Query 1: Minor language revision (e.g. “words counts” instead of word count, “cerebral infarction” instead of cerebral ischemia, Table 1 (no period at the end of each phrase…). Please revise.

: We have made the corrections.

“glaucomatous optic neuropathy (GON)”

# Query 2: Table 1. Exclusion criteria, uncontrolled IOP > 21 despite medication. Newly diagnosed glaucoma is the main inclusion criteria highlighted by the authors, therefore this exclusion criteria seems to be redundant. Please explain.

: We agree with the reviewer’s comments. “Newly diagnosed glaucoma” in the inclusion criteria means that treatment-naïve patients at baseline will be enrolled in the study. In contrast, the exclusion criterion is patients with higher IOP despite being under maximal tolerated medical therapy during follow-up. The authors judged this to be the dropout criterion during follow-up rather than the baseline exclusion criteria. Therefore, we have deleted the related exclusion criteria part, and the contents regarding the ‘End points’ part were reinforced. 

(End Points in Methods, Page 10)

“Endpoints requiring discontinuation in the study included: treated IOP >21 mmHg despite maximal tolerated medical therapy on two successive occasions (safety end point); glaucoma surgery or glaucoma laser treatment (laser iridotomy, laser iridoplasty, laser trabeculoplasty) (therapeutic end point); pregnancy or lactation during follow-up; missing two or more consecutive questionnaire responses. Data up to this point were included in the analysis, but discontinued patients were no longer followed as participants.”

# Query 3: Patients with uncomplicated cataract surgery can be included in the study? There seems to be a contradiction, a following paragraph states that a history of cataract during the previous 6 months is an exclusion criteria. Please, explain.

: Thank you for your important comment. To reduce misunderstanding, we have modified the contents further.

(Subjects in Methods, Page 6)

“ Patients with a history of intraocular surgery (other than uncomplicated cataract extraction more than 6 months previously)”

Table 1. Inclusion and exclusion criteria

Exclusion criteria

• Previous intraocular surgery (other than uncomplicated cataract extraction more than 6 months previously)

# Query 4: Ophthalmologic assessment. The authors should provide the model of each device used in the study. All the participant centers should use the same model of each device.

: We thank the reviewer for raising this valid point. According to the reviewer’s comment, we indicated the model and manufacturer of each device. It was confirmed that the main device (optic disc, red-free fundus photography, OCT, and HVF) used to evaluate glaucoma progression had the same manufacturer at the four institutions. The related contents are specified in the Methods part.

(Ophthalmologic Assessments in Methods, Page 7)

“All enrolled patients will undergo a comprehensive ophthalmological examination, including assessment of best-corrected visual acuity (BCVA), slit lamp biomicroscopy, intraocular pressure (IOP) measurement by Goldmann applanation tonometry (Haag-Streit), anterior chamber angle measurement by gonioscopy, stereoscopic examination of the optic disc, red-free fundus photography (TRC-50IX; Topcon Corporation), central corneal thickness measurement with ultrasound pachymetry, and axial length measurement by optical biometer. RNFL thickness will be measured by Cirrus optical coherence tomography (OCT) (Carl Zeiss Meditec, Dublin, CA, USA), and Humphrey visual field (HVF) (Humphrey Field Analyzer II; 24–2 Swedish Interactive Threshold Algorithm; Carl Zeiss Meditec). Testing will be performed using Swedish Interactive Threshold Algorithm standard 24-2 perimetry. The same optic disc, red-free fundus photography, OCT and HVF models will be used across all of the research institutions.”

# Query 5: “Glaucoma progression will be determined by observing structural and functional changes. The markers to be included are rim notching or thinning, cup contour alteration, optic disc vasculature change, and increased cup-to-disc ratio. RNFL progression will be identified by defect depth/width and new defects”. How will these changes be reported by each investigator?

: The authors appreciate the reviewer’s comment. The clinical interpretations of VF, optic nerve head appearance and overall disease stability were to be collected during FU. It is displayed in the medical records during follow-up observation, the evaluation and report are made after the study is completed, and the clinician conducts the evaluation blinded to patient information. We have added the relevant contents to the Methods part.

(Glaucoma Progression Assessment in Methods, Page 12)

“During the research period, data are accumulated without any evaluation of progress. The final trial dataset will be blinded to any identifying participant information and uploaded to an open-access data repository. The analysis probably will be completed after the 2-year study period. At the end of the study, each participant’s clinical data will be assessed for progress.” 

# Query 6. The authors mentioned other cross-sectional and retrospective studies about the subject. Those studies included a very wide number of cases (6.760 to more than 4 millions…). The authors plan to include 300 patients, but I think they didn´t provide the statistical method to calculate the sample size.

: Thank you for your comments and excellent suggestions. We have added the statistical explanation for calculation of sample size and provided references for these assumed data.

(Sample Size Calculation in Methods, Page 21)

“The sample size was based on the outcome of the Early Manifest Glaucoma Trial (EMGT):29 the glaucoma progression rates at 24 months were 20% in the untreated group and 10% in the treated group, and IOP reduction was 25% in the treated group. In this study, baseline-target IOP was set at 20 to 30% IOP reduction, depending on baseline IOP. Based on 90% power (1–β) at a 2-sided error α = 0.05 to detect the difference between 20 and 10% in incident progression over a 24-month follow-up, 125 patients will be needed in each arm (250 total). Allowing for 20% attrition over the study period, 300 patients will need to be recruited for the study.”

Reviewer #2

# General comments:

Association Between Psychological Vulnerability and Glaucoma Progression: Protocol for a Multicenter Prospective Cohort Study in South Korea

The authors aim to study a very important facet of glaucoma which is often overlooked: the psychologic impact of this disease on our patients. Importantly, they also aim to examine the role of psychologic diagnoses in glaucoma worsening via its role in medication noncompliance. I commend the authors for identifying this area of much needed research and designing a study to gather this information.

: We greatly appreciate the reviewer's advice and suggestions. The manuscript has been revised accordingly. Modifications and supplements were made to the study design. The detailed results of the revision are outlined in our responses to the respective comments.

# Query 1. A large part of psychologic vulnerability and glaucoma could be impacted by the physician patient relationship. Is there standard way patients are told about their disease? Is the education provided standardized for each patient enrolled? Some physicians resort to scare tactics during diagnosis which causes increased anxiety in patients and this could influence results and the patient’s reaction to their diagnosis.

: We totally agree with your suggestion. Each of the clinicians at the four institution may have different styles in the process of delivering the diagnosis. Through prior education, we will make efforts to ensure that the explanations in the diagnostic process are relatively unified and standardized. We have added a relevant sentence to the Methods section.

(Study Design in Methods, Page 5)

“According to prior training, clinicians at each institution will try to use a standardized method of delivering the diagnosis of glaucoma.”

# Query 2: Methods: subjects

What is your definition of a reliable visual field test, this should be defined in the protocol.

: As per the reviewer’s suggestion, the reliable VF test criterion has been added to the Methods section.

(Subjects in Methods, Page 6)

“VF defects will be confirmed based on reliable tests (fixation loss rate ≤ 20%, false positive and false negative error rates ≤ 25%).”

# Query 3: Will background demographics for each patient be collected? This study will need to control for age, sex, socioeconomic status, education level, transportation, employment status, social support network etc as these may play an important role in psychologic response to diagnosis. I assume this will be collected but it is not explicitly stated.

: We appreciate the reviewer’s comment. The authors agree that age, sex, socioeconomic status, transportation, social support network, etc. can affect a patient's anxiety and access to medical care. Information on the confounding factors has been supplemented in the Methods section.

(Statistical Analysis Plan in Methods, Page 13)

“All covariates will be collected at baseline. Patients will be grouped into five categories based on age: <40, 40‒49, 50‒59, 60‒69, and ≥70 years. Sex will be categorized as male or female. Health insurance will be categorized based on the Korean National Health Insurance System (KNHIS) as employee, self-employed individual, or medical aid beneficiary. Residence regions will be grouped into metropolitan area (Seoul, Busan, Daegu, Incheon, Daejeon, Gwangju, Ulsan) or local province (Gangwon, Gyeonggi, Chungcheongbuk, Chungcheongnam, Sejong, Gyeongsangbuk, Gyeongsangnam, Jeollabuk, Jeollanam, Jeju).”

# Query 3: Is there a standard for treatment? How can you control for inadequate treatment causing progression versus medication noncompliance? Will patients all be started on similar therapies? 

: In this study, participants will be starting on similar medical treatments. The standard treatment in this protocol is topical antiglaucoma medications. Patients will be treated with a suitable line of medication according to the initial IOP and maintain the target IOP after reaching it. Monotherapy or combination (prostaglandin analogues, β-blockers, carbonic anhydrase inhibitors, α-adrenergics) will be used as first-line drugs. This study will set the target IOP as a therapeutic guideline. According to clinical judgment, topical antiglaucoma medication will be added when the IOP reduction is insufficient relative to the target IOP. In this study, cases with low medication adherence will be classified and analyzed as low adherence. Clinically, it is difficult to distinguish between low treatment efficacy and low medication adherence, but we plan to encourage clinicians to use the medication when low adherence is identified.

We have added relevant sentences to the ‘Medication Adherence’ paragraph.

(Medication Adherence in Methods, Page 9)

“In this study, a target IOP showing at least a 20-30% reduction from the initial pressure will be set and clinicians at each institution will use it as a guideline for lowering of IOP.26 Clinicians may adjust medications if IOP control is insufficient or if glaucoma progression occurs during follow-up.”

# Query 4. # Is there a standard percentage of IOP lowering that will be targeted as a guide for initial treatment or is this left to the discretion of the treating provider?

: It is generally assumed that aiming to achieve a target IOP showing at least a 20-30% reduction from the initial pressure at which damage occurred is a helpful starting point. In this study, the target IOP will also be set, so clinicians at each institution can use it as a guideline for lowering of IOP. The target IOP should be reassessed periodically and lowered if progression, optic nerve hemorrhage, or increased risk factors occur. In this study, if the IOP reduction is insufficient relative to the target IOP level, the topical medication will changed or another added; this will be left to the discretion of the treatment provider. The relevant information is summarized as follows.

(Medication Adherence in Methods, Page 9)

“In this study, a target IOP showing at least a 20-30% reduction from the initial pressure will be set, and clinicians at each institution will use it as a guideline for lowering of IOP.26 Clinicians may adjust medications if IOP control is insufficient or if glaucoma progression occurs during follow-up. The target IOP will be reassessed periodically and lowered if progression, optic nerve hemorrhage, or increase in risk factors occurs.”

# Query 5 Medication Adherence:

Using patient reported adherence is a very poor way to assess adherence. Most patients overestimate their adherence to medication. Pharmacy refill data or devices which track adherence are a much more reliable measure of medication adherence in this technological climate. How will patients keep track of this? Many patients do not recall they have missed doses?.

: We agree that using patient-reported adherence is not an effective and reliable standard. Particularly for disposable preservative-free drugs, these dosage refill data seem reliable for evaluation of medication adherence. Still, in the case of bottle formulations, there is a large individual difference in usage. In the case of old age for example, a large amount of drug is lost due to poor aiming. Therefore, adherence measurement through this refill data also seems to have limitations. 

Ultimately, this study decided to adopt the questionnaire method as originally planned, since medication adherence needs to be evaluated quantitatively. But reflecting the reviewer's opinion, this part was added to the research limitations part.

(Limitations in Discussion, Page 14)

“Third, using patient-reported adherence might not be a reliable way to assess adherence, as most patients have a tendency to overestimate their adherence to medication.”

# Query 6 : Analysis of Glaucoma Progression

In newly diagnosed glaucoma patients, tracking patients over 2 years may not provide enough data (with 4 visual fields? i.e. one every 6 months) to determine progression. It will give an idea but often there is a learning curve in field testing.

: The authors deeply appreciate the reviewer’s drawing our attention to this issue. Accordingly, we tried to collect enough data from a sufficient number of reliable visual field tests. In detail, at the time of initial evaluation, consecutive visual field tests will be performed on a single day or on another day to secure two or more reliable baseline VF tests. In addition, a visual field test will be performed additionally at 3 months. The related contents have been modified, and the study flow diagram (Figure 1) has been revised as well.

(Ophthalmologic Assessments in Methods, Page 8)

“Visual acuity testing, IOP measurement, and optic disc examination will be conducted at each visit, while OCT, HVF, optic disc, and red-free photography will be evaluated at 3 months and every 6 months (Figure 1).”

Reviewer #3

# General comments:

The idea of this protocol is excellent and nowadays very popular. Also the paper contains interesting findings, however, there are following issues to be addressed:

: The authors truly appreciate the reviewer’s supportive comments. The manuscript has been revised accordingly. We have added a further explanation of the study design. In addition, some references were replaced with relatively recent studies. The details on the revision are outlined in our responses to the respective comments.

# Query 1. Only patients with open-angle glaucoma (primary open-angle glaucoma [POAG] and normal-tension glaucoma [NTG]) will be included in the study? And if so, why?

: The authors appreciate the reviewer’s concern. If there is an event such as an angle-closure attack, the psychological state fluctuates. Also, because surgical procedures are more frequent than medication treatments, it may be limited in assessing drug compliance. For the above reasons, this study included OAG except for angle-closure glaucoma.

# Query 2. Why only one eye per patient will be included, or if both eyes are eligible, the eye with the more severe mean deviation (MD) value from the visual field test?

: Thank you for your excellent comment. Another reviewer gave a similar comment and suggested that patients with monocular and binocular glaucoma may have different levels of anxiety. Additionally, in bilateral glaucoma, it can be controversial whether it is the worse eye or the better eye that has a greater impact on the patient's psychological state. For the above reasons, in this study, even in the case of binocular glaucoma, only one eye was used for analysis, and a random selection method was adopted for the selection of one eye.

(Subjects in Methods, Page 5)

“Only one eye per patient will be included, and if both eyes meet the eligibility criteria, one eye will be randomly included.”

# Query 3. Whether experience of a personal trauma such as death of a loved one, job loss or recent divorce will be taken into account as an exclusion criterion?

: We appreciate the reviewer’s comment. The authors fully agree that experience of a personal trauma such as death of a loved one, job loss or recent divorce could affect the psychological state. Therefore, we added this point to the exclusion criteria. In addition, since such an event may be related to personal privacy, we will add it to the patient's prior informed consent so that the patients themselves can check whether he or she is eligible.

Table 1. Inclusion and exclusion criteria

Exclusion criteria

• Experience of a personal trauma such as death of a loved one, job loss or divorce within the previous 6 months.

# Query 4. Whether psychiatry specialist will be included in the study and for the interpretation of the results?

: Thank you for raising this important point. We will also consult on the interpretation of the psychiatric questionnaire results. In addition, in the case of a severe depression/anxiety/stress scale, we plan to see a psychiatric specialist in consultation with the patient. The paragraphs specified below are the related content.

(Anxiety/Depression/Stress Scales in Methods, Page 8)

“Participants will be able to complete the questionnaires at the clinic or verbally if necessary. Consultation with a psychiatrist will facilitate interpretation of the survey results.”

(Psychiatric management in Methods, Page 9)

“Additional visits to a psychiatry specialist will be scheduled for a patient if a patient’s GAD-7, PHQ-9, or PSS indicating severe stage. Further evaluation and treatment advice will be made by psychiatrists. Treatment of depression or anxiety is based on the patient’s own will and is not mandatory. Subsequently, antidepressant or antianxiety treatment information will be recorded in Case Report Forms (CRFs).”

# Query 5: Why were the scales: the Generalized Anxiety Disorder Assessment (GAD-7), the Patient Health Questionnaire (PHQ-9) chosen to assess anxiety and depression?

: Thank you for your question. Since the GAD-7, PHQ-9, and PSS-10 questionnaires are widely used and their reliability and accuracy have been proven in other psychiatric evaluation studies, those scales were adopted. The authors considered that the explanation for why these scales were chosen to assess anxiety, depression, and stress was insufficient, and thus, the rationale for their selection was supplemented in the revision process.

(Anxiety/Depression/Stress Scales in Methods, Page 8)

“The GAD-7, PHQ-9, and PSS-10 scales are short and understandable screening instruments that are useful for detecting and assessing severity of symptoms in clinical and research settings. In addition, these scales have been formally validated against diagnostic clinical interviews to establish the sensitivity, specificity and positive and negative predictive values of cut-off scores and are used widely in clinical research.15-17”

# Query 6 : Out of 25 references, only three are 5 years old.

: According to the reviewer’s suggestion, the reference was changed to the latest study as much as possible. The following are the references published within the last 5 years that have been added.

<References>

4. Montana CL, Bhorade AM. Glaucoma and quality of life: fall and driving risk. Current opinion in ophthalmology. 2018;29(2):135-140. 

5. Liu WW, Shalaby WS, Shiuey EJ, et al. Correlation between Central Visual Field Defects and Stereopsis in Patients with Early-to-Moderate Visual Field Loss. Ophthalmology Glaucoma. 2023;

9. Groff ML, Choi B, Lin T, Mcllraith I, Hutnik C, Malvankar-Mehta MS. Anxiety, depression, and sleep-related outcomes of glaucoma patients: systematic review and meta-analysis. Canadian Journal of Ophthalmology. 2022;

11. Stamatiou M-E, Kazantzis D, Theodossiadis P, Chatziralli I. Depression in glaucoma patients: a review of the literature. Taylor & Francis; 2022:29-35.

12. Sanchez FG, Mansberger SL, Newman-Casey PA. Predicting adherence with the glaucoma treatment compliance assessment tool. Journal of glaucoma. 2020;29(11):1017. 

Reviewer #4

# General comments:

This article reports on the study protocol of a prospective multicenter cohort study designed to investigate the impact of psychological states such as anxiety, depression, and stress on both medication adherence and glaucoma progression. This cohort study, if successfully conducted and completed, will contribute to the understanding of changes in the psychological state of glaucoma patients during the course of treatment and provide valuable clinical information for the psychiatric management of patients with glaucoma. However, the study protocol reported in this article is incomplete and needs to be revised (see comments below).:

: The authors truly appreciate the reviewer’s supportive comments. Thanks to the reviewer's excellent advice, the design of this study has become much more logical and organized through revision. The detailed results of the revision are outlined in our responses to the respective comments.

# Query 1. Ethics Statement:

The authors provide two OSF registration numbers in this manuscript. Please confirm which one is correct. Is it DOI 10.17605/OSF.IO/AWTEC (mentioned in the manuscript information) or DOI 10.17605/OSF.IO/DA2F4 (mentioned in the main text)?

: The authors are sorry for the mistake. We have made the DOI consistent throughout the manuscript.

OSF Registration Number

DOI 10.17605/OSF.IO/DA2F4

(Ethics and Declaration in Methods, Page 5)

“The study protocol has been registered in the Open Science Framework (OSF) under registration number DOI 10.17605/OSF.IO/DA2F4.”

# Query 2. Subjects

① “…if both eyes are eligible, the eye with the more severe mean deviation (MD) value from the visual field test will be selected.”

I would like to know whether it is the worse eye or the better eye that has a greater impact on the patient's psychological state. If a participant with glaucoma reaches legal blindness in one eye and still has better vision in the other eye, say 20/30, he/she may be in two states of mind, fearing deterioration in the better-sighted eye or, living unaffected because he/she still has one eye with adequate vision. I wonder if the worse eye or the better eye could have a greater impact on the patient's psychological states. If the conclusions in this aspect are controversial, I would suggest that if both eyes meet the eligibility criteria, one eye be randomly included.

:Thank you for your excellent comments. We totally agree with your suggestion. Another reviewer gave similar comments and suggested that patients with monocular and binocular glaucoma may have different levels of anxiety. Additionally, in bilateral glaucoma, it can be controversial whether it is the worse eye or the better eye that has a greater impact on the patient's psychological state. Accordingly, even in the case of binocular glaucoma, only one eye was used for analysis, and a random selection method was adopted for the selection of one eye.

(Subjects in Methods, Page 5)

“Only one eye per patient will be included, and if both eyes meet the eligibility criteria, one eye will be randomly included.”

# Query 2. Subjects

② “VF defects will be confirmed based on specific criteria, and reliable tests will be conducted.” Please add the definition of glaucomatous VF damage and the criteria of reliable tests.

: According to reviewer’s suggestion, we have added the definition of glaucomatous VF damage and the criteria of reliable tests.

(Subjects in Methods, Page 6)

Glaucomatous eyes were defined as those showing glaucomatous optic disc neuropathy (e.g., notching, neuroretinal rim thinning, and/or RNFL defects) and corresponding glaucomatous VF defects, as confirmed by at least two consecutive VF examinations. Glaucomatous VF defects were defined as a cluster of ≥ 3 points with P < 0.05 on the pattern deviation map in at least one hemifield, including ≥ 1 point with P < 0.01; a pattern standard deviation of P < 0.05; or glaucoma hemifield test result outside the normal limits.

# Query 2. Subjects

• Open-angle glaucoma characterized by open angles, glaucomatous optic nerve damage, and confirmed visual field damage” Please maintain consistency of statements. Is secondary open-angle glaucoma excluded?

: Thank you for your valuable comments. Secondary open-angle glaucoma, such as pseudoexfoliation glaucoma and steroid-induced glaucoma, will be included in this study. Therefore, the related inclusion criteria have been changed as follows.

(Subjects in Methods, Page 6)

“Patients will have to meet specific criteria for open-angle glaucoma, including glaucomatous optic disc changes, retinal nerve fiber layer (RNFL) defects, glaucomatous visual field (VF) defects, and open angle as confirmed by gonioscopic examination.”

# Query 2. Subjects 

③ “Exclusion criteria:

• Uncontrolled intraocular pressure (> 21 mmHg) despite medication”

Since the participants are all newly diagnosed with glaucoma, it will take at least some time to see if the IOP can be controlled with medication alone. What is the approximate length of this period? Will the baseline data be collected before or after this period?

: We fully appreciate the excellent questions. “Newly diagnosed glaucoma” in the inclusion criteria means that treatment-naïve patients at baseline will be enrolled in the study. Whereas the exclusion criterion is patients with uncontrolled IOP despite being under maximal tolerated medical therapy during follow-up. The authors judged this to be the dropout criterion during follow-up rather than the baseline exclusion criteria. Therefore, the related exclusion criteria part was deleted, and the contents regarding the End point part were reinforced.

(End Points in Methods, Page 10)

“Endpoints requiring discontinuation from the study included: treated IOP >21 mmHg despite maximal tolerated medical therapy on 2 successive occasions (safety end point),”

④ Is undergoing glaucoma surgery an indicator of glaucoma progression, or the endpoint of this study?

: Thank you for your incisive question. In this study, undergoing glaucoma surgery or laser treatment during the follow-up was considered to be the endpoint criterion. In this case, we plan to participate in the study only up to the point of decision for surgery or laser procedure. We have added related content to the paragraph regarding the End points.

(End Points in Methods, Page 10)

“Endpoints requiring discontinuation from the study included: treated IOP >21 mmHg despite maximal tolerated medical therapy on 2 successive occasions (safety end point); glaucoma surgery or glaucoma laser treatment (laser iridotomy, laser iridoplasty, laser trabeculoplasty) (therapeutic end point); pregnancy or lactation during follow-up. Data up to this point were included in the analysis, but discontinued patients were no longer followed as participants.”

⑤ Study exclusion criteria include pregnancy and lactation. During the enrollment process, will participants be informed of the need to use contraception or breastfeed during the 2-year study?

: In the baseline enrollment, pregnant or lactating patients will be excluded. However, prohibiting pregnancy or lactation during the follow-up would be an unethical policy. There are no binding provisions in this study. However, a participant could become pregnant or need breastfeeding during the study period. In this case, glaucoma medications may be withdrawn, and the conduct of the scheduled test and regular follow-ups will be restricted. Therefore, these cases will be set as endpoints in this study. The exclusion criteria and endpoint issues have been modified to reduce this misunderstanding.

 Table 1. Inclusion and exclusion criteria

Exclusion criteria

• Pregnancy or lactation at enrollment. 

(End Points in Methods, Page 10)

“End points requiring discontinuation from the study included: treated IOP >21 mmHg despite maximal tolerated medical therapy on two successive occasions (safety end point); glaucoma surgery or glaucoma laser treatment (laser iridotomy, laser iridoplasty, laser trabeculoplasty) (therapeutic end point); pregnancy or lactation during follow-up, and missing two or more consecutive questionnaire responses. Data up to this point were included in the analysis, but discontinued patients were no longer followed as participants.”

⑥ If a participant has experienced a vital event within the 6 months or during the study period, e.g., birth, death, marriage/civil partnership, divorce, adoption, would he or she be excluded? These life events are known to affect a person's social behavior and mental state. How will the researcher investigator deal with the impact of these events on the study results?

: We appreciate the reviewer’s question. The authors fully agree that experience of a personal trauma such as death of a loved one, job loss or recent divorce could affect the psychological state. Therefore, we added this point to the exclusion criteria. In addition, since such an event may be related to personal privacy, we will add it to the patient's prior informed consent so that the patients themselves can check whether he or she is eligible.

Table 1. Inclusion and exclusion criteria

Exclusion criteria

• Experience of a personal trauma such as death of a loved one, job loss or divorce within the previous 6 months.

⑦ I was curious that high myopia may also lead to some structural changes in the fundus similar to POAG, especially in the optic disc area. Based on the long follow-up time proposed in this study, I suggest that diopter or axial length restrictions be added to the exclusion criteria. If not, please give a reason that can be identified.

: Thank you for your comment and suggestion. High myopia may have limitations in quantitative analyses including OCT analysis. Therefore, high myopia patients were excluded from this study. We have added the items to Table 1.

Table 1. Inclusion and exclusion criteria

Exclusion criteria

• High myopia (defined as spherical equivalent > −6.0 diopters)

⑧ Will a participant be treated by the same physician throughout the follow-up period?

: All participants will receive management and follow-up by a single physician at each institution. We have added that point to the related content in the Methods section. 

(Study Design in Methods, Page 5)

“All participants will be treated by a single physician throughout the follow-up period at each institution.”

⑨ Please indicate if the participant will receive any grants, bonuses or incentives.

: There are no grants, bonuses or incentives for study participation in this study. Relevant information has been added to the text.

(Subjects in Methods, Page 6)

“No subjects will receive any grants, bonuses or incentives for participating in the study.”

# Query 4. Trial design

① Please state whether this prospective study involves blinding:

a. For example, only the study designers/allocators knew the patients' initial mental scores and the physicians were blinded to the scores. This may have allowed physicians to avoid giving more aggressive treatment to patients with lower scores.

b. Are the allocators who decide on the need for psychiatric intervention in patients independent and blinded to the stage of glaucoma?

c. Are the investigator(s) performing the statistical analyses blinded?

d. If this study is not blinded, what means will the investigator use to reduce the associated bias?

The authors appreciate the excellent comments and suggestion. The authors agree that these points are essential to reducing research bias. As per the reviewer’s advice, under the ‘Masked and Blinded Evaluation’ subheading, we have added the following contents to the Methods section.

(Masked and Blinded Evaluation in Methods, Page 12)

The masked and blinded evaluation for minimization of bias can be summarized as follows:

 Masked technicians who obtain the IOP measurements are not masked to the patient's participation in the study.

 Only the study designers/allocators know the patients' initial psychological scores, and the physicians are blinded to the scores.

 Allocators who decide on the need for psychiatric intervention in patients are independent and blinded to the stage of glaucoma.

 Investigators blindly perform the statistical analyses.

② Given the long follow-up proposed in this study, how will missing responses be handled?

: Thank you for the comments. We will try to adjust the date of the outpatient visit or make an effort not to miss a response by the allocator. Nevertheless, in the case of consecutive omissions in important survey evaluations, withdrawal is processed through consultation among researchers (End points). Relevant contents were added in the End points part.

(End Points in Methods, Page 10)

“End points requiring discontinuation from the study included: treated IOP >21 mmHg despite maximal tolerated medical therapy on two successive occasions (safety end point); glaucoma surgery or glaucoma laser treatment (laser iridotomy, laser iridoplasty, laser trabeculoplasty) (therapeutic end point); pregnancy or lactation during follow-up, and missing two or more consecutive questionnaire responses. Data up to this point were included in the analysis, but discontinued patients were no longer followed as participants.”

③ Does this study include a control population?

: There is no plan to specifically include a control population in this study.

# Query 5. Anxiety/Depression/Stress Scales

I suggest that the questionnaires also be administered at the 6-month and 18-month follow-up visits. The Kübler-Ross model has five successive stages. If it is true, as the authors suggest, that the mental status of participants after glaucoma diagnosis is similar to the Kübler-Ross model, then administering the questionnaire only at baseline, 1-year, and 2-year follow-up visits may be insufficient.

: The authors totally agree with the reviewer’s suggestion. During the revision process, the schedule was changed so that questionnaires will also be administered at the 6-month and 18-month follow-up visits.

(Anxiety/Depression/Stress Scales, Page 10)

To that end, self-rating questionnaires will be administered at baseline and at every 6-month follow-up visit (Figure 1).

(Revised Figure 1.)

# Query 6. Analysis of Psychiatric Scales and Glaucoma Progression

① Please state the primary, secondary and exploratory objectives (if any) and safety objectives of this study.

: The authors really appreciate the reviewer’s drawing this to our attention. We summarized the primary and secondary objectives of this study during the revision. Since this trial is not a study evaluating procedures or drug administration to patients, specific safety objectives do not exist.

(Analysis of Psychiatric Scales and Glaucoma Progression, Page 10)

Primary Objectives. The main objective of this study is to investigate the relationship between an individual's psychiatric state (as measured on anxiety/depression/stress scales) and progression of glaucoma. We also aim to analyze how anxiety, depression and stress levels affect medication adherence during glaucoma treatment.

Secondary Objectives. The secondary objectives are (1) to evaluate speed of glaucomatous progression according to psychological scales, (2) to identify psychological factors for medication non-adherence, and (3) to identify ophthalmic factors for psychological deterioration.

② The first and second points of the study results appear to be repetitive.

: The authors agree with reviewer’s point. We have eliminated the repetitive contents.

(Analysis of Psychiatric Scales and Glaucoma Progression, Page 11)

The detailed study-outcomes to be investigated can be summarized as follows:

 The association between the initial anxiety/depression/stress scales and the severity of glaucoma at the time of diagnosis.

 The correlation between the initial anxiety/depression/stress scales and the glaucoma progression rate or medication adherence.

 The longitudinal change pattern of psychiatric scale between progression and non-progression group.

 Changes in medication adherence and psychological scale according to follow-up.

 Risk factors for psychological deterioration and medication non-adherence in glaucoma patients.

③ The described study outcomes focus on the correlation between the initial scale and glaucoma. There is a lack of study outcomes on the dynamic changes during the course of the study.

: Thank you for your important comments. We have added the study outcomes regarding the dynamic change for each parameter that is repeatedly measured at follow-up visits. The items that have been added are summarized below.

(Analysis of Psychiatric Scales and Glaucoma Progression, Page 11)

 The longitudinal change pattern of psychiatric scale between progression and non-progression group.

 Changes in medication adherence and psychological scale according to follow-up.

 Risk factors for psychological deterioration and medication non-adherence in glaucoma patients.

# Query 7. Glaucoma Progression Assessment

“Glaucoma progression will be determined by observing structural and functional changes. The markers to be included are rim notching or thinning, cup contour alteration, optic disc vasculature change, and increased cup-to-disc ratio.”

① Are changes in the cup-to-disc ratio based on the subjective judgment of the clinician or the objective analysis of the testing instrument?

: Cup-to-disc-ratio (CDR) is evaluated by using disc photography. At this time, whether the CDR increases or not may be somewhat subjective. However, CDR change can be comprehensively evaluated by checking the RNFL profile (red-free photo and OCT). The authors consider that these structural evaluations are judged to be relatively accurate. Relevant information was added to the paragraph on “Glaucoma Progression Assessment”.

(Glaucoma Progression Assessment in Methods, Page 12)

“Progressive pathologic changes in the optic disc will be evaluated by comparison of serial color disc photographs. RNFL progression will be identified by defect depth/width and new defects. Optic disc features are comprehensively evaluated by checking the related RNFL profile (red-free photo and OCT).”

② Please subpoint the conditions that are considered glaucoma progression, as this appears to be the endpoint of this study. 

: In this study, glaucoma progression is not set as an endpoint; in this case, the topical medication will be added under the clinician's judgment. However, when IOP is not controlled despite maximal tolerating medical treatment or receiving glaucoma surgery or laser procedures, the study observation will be terminated by setting endpoint criteria. The following related endpoint contents were added during the revision process.

(End Points in Methods, Page 10)

“End points requiring discontinuation from the study included: treated IOP >21 mmHg despite maximal tolerated medical therapy on two successive occasions (safety end point); glaucoma surgery or glaucoma laser treatment (laser iridotomy, laser iridoplasty, laser trabeculoplasty) (therapeutic end point); pregnancy or lactation during follow-up; missing two or more consecutive questionnaire responses. Data up to this point were included in the analysis, but discontinued patients were no longer followed as participants.”

# Query 8. Statistical Analysis Plan and Handling Bias

① “This multicenter study estimates that about 300 participants would be needed, assuming the accuracy and reliability of the questionnaire to be 80% with the target number of sample size.”

Please explain the sample size calculation. What is the assumption? Is the test one- or two-sided? How much statistical power does the sample size provide? And please provide references for these assumed data.

: Thank you for your questions and excellent suggestion. We have added a statistical explanation for the calculation of sample size and provided references for these assumed data.

(Sample Size Calculation in Methods, Page 12)

“The sample size was based on the outcome of the Early Manifest Glaucoma Trial (EMGT):29 the progression rates at 24 months were 20% in the untreated group and 10% in the treated group, and IOP reduction was 25% in the treated group. In this study, baseline-target IOP was set at 20 to 30% IOP reduction, depending on baseline IOP. Based on 90% power (1–β) at a 2-sided error α = 0.05 to detect the difference between 20 and 10% in incident progression over a 24-month follow-up, 125 patients will be needed in each arm (250 total). Allowing for 20% attrition over the study period, 300 patients will need to be recruited to the study.”

② Jeju-do has a much smaller population than the Seoul area and Chungcheongnam-do. And two of the four hospitals are located in the Seoul area. What is the expected proportion of participants to be enrolled in each of the four centers? And how can uneven enrollment be avoided?

: Thank you for your valuable comments. The selection of the four multicenters in this study reflected geographical factors. Considering geographical distribution, Seoul National University Hospital and Hallym University Hospital reflect the metropolitan areas. And Chungnam National University Sejong Hospital and Jeju National University Hospital represent local provinces. The total subjects will be enrolled equally from the four institutions. As per the reviewer’s comments, Jeju Island has a small population, but Jeju Hospital, a tertiary medical institution, is expected to have no clinical difficulties in enrolling participants. Consistent with the reviewer's opinion, the geographical location and demographics of the four institutions are uneven enrollment to be a fully representative sample. We will investigate the patients’ addresses later and adjust them as covariates during the analysis.

(Statistical Analysis Plan in Methods, Page 13)

“All covariates will be collected at the baseline. Patients will be grouped into five categories based on age: <40, 40‒49, 50‒59, 60‒69, and ≥70 years. Sex will be categorized as male or female. Health insurance will be categorized based on the Korean National Health Insurance System as employees, self-employed individuals, or medical aid beneficiaries. Residence regions will be grouped into metropolitan area (Seoul, Busan, Daegu, Incheon, Daejeon, Gwangju, Ulsan) or local province (Gangwon, Gyeonggi, Chungcheongbuk, Chungcheongnam, Sejong, Gyeongsangbuk, Gyeongsangnam, Jeollabuk, Jeollanam, Jeju).”

③ Please briefly describe how the researcher will reduce the biases. 

: Although it overlaps with what the author said earlier, we will try to reduce the bias through various blind evaluations. Under the ‘Masked and Blinded Evaluation’ subheading, we have added contents to the Methods section.

(Masked and Blinded Evaluation in Methods, Page 12)

The masked and blinded evaluation to reduce bias can be summarized as follows:

 Masked technicians who obtain the IOP measurements are not masked to the patient's participation in the study.

 Only the study designers/allocators know the patients' initial psychological scores, and the physicians are blinded to the scores.

 Allocators who decide on the need for psychiatric intervention in patients are independent and blinded to the stage of glaucoma.

 Investigators blindly perform the statistical analyses.

# Query 9. Ethics and Declaration Please place this section before or merge it with the Study Design section

: As per reviewer’s suggestion, the ‘Ethics and Declaration’ section was placed in front of the ‘Study Design’ section.

---

## [Decision Letter · Decision Letter 1]

25 Sep 2023

Association Between Psychological Vulnerability and Glaucoma Progression: Protocol for a Multicenter Prospective Cohort Study in South Korea

PONE-D-23-18243R1

Dear Dr. Kim,

We’re pleased to inform you that your manuscript has been judged scientifically suitable for publication and will be formally accepted for publication once it meets all outstanding technical requirements.

Kind regards,

Alvaro Jose Mejia-Vergara, MD

Academic Editor

PLOS ONE

Additional Editor Comments (optional):

Great job!

one of the reviewers still has a question about the cut-off point for the Visual Fields reliability values, but I don't see them being either against already published data or usual clinical practice. This manuscript will be accepted.

Reviewers' comments:

Reviewer's Responses to Questions

**Comments to the Author**

1. Does the manuscript provide a valid rationale for the proposed study, with clearly identified and justified research questions?

Reviewer #1: Yes

Reviewer #2: Yes

Reviewer #3: Yes

Reviewer #4: Yes

2. Is the protocol technically sound and planned in a manner that will lead to a meaningful outcome and allow testing the stated hypotheses?

Reviewer #1: Yes

Reviewer #2: Yes

Reviewer #3: Yes

Reviewer #4: Yes

3. Is the methodology feasible and described in sufficient detail to allow the work to be replicable?

Reviewer #1: Yes

Reviewer #2: Yes

Reviewer #3: Yes

Reviewer #4: Yes

4. Have the authors described where all data underlying the findings will be made available when the study is complete?

Reviewer #1: Yes

Reviewer #2: Yes

Reviewer #3: Yes

Reviewer #4: Yes

5. Is the manuscript presented in an intelligible fashion and written in standard English?

Reviewer #1: Yes

Reviewer #2: Yes

Reviewer #3: Yes

Reviewer #4: Yes

6. Review Comments to the Author

You may also provide optional suggestions and comments to authors that they might find helpful in planning their study.

Reviewer #1: Thank you for your work about this very important topic and for the improvements made to the article.

Regards

Reviewer #2: The authors have addressed a majority of the concerns mentioned in the previous review.

A few follow up questions as follows:

1. Regarding the visual field reliability metrics, were these arbitrarily decided or based on previous publications? The false negative and false positive rate allowances are quite high and would qualify as unreliable by most glaucoma practitioners even with very lax standards. Particularly the false positive rate which could artificially improve the MD by up to 2 dB.

2. The level of education should be recorded for the included subjects as this could be a large confounder and also play a role in patient understanding of disease.

Reviewer #3: The authors responded to all the reviewers' comments and suggestions, so I have no additional comments to authors.

Reviewer #4: After careful revisions by the authors, the manuscript has improved significantly and could be accepted for publication.

7. PLOS authors have the option to publish the peer review history of their article (what does this mean?). If published, this will include your full peer review and any attached files.

Reviewer #1: No

Reviewer #2: **Yes: **Elyse Mcglumphy

Reviewer #3: No

Reviewer #4: No

---

## [Editor Report · Acceptance letter]

28 Sep 2023

PONE-D-23-18243R1 

Association Between Psychological Vulnerability and Glaucoma Progression: Protocol for a Multicenter Prospective Cohort Study in South Korea 

Dear Dr. Kim:

I'm pleased to inform you that your manuscript has been deemed suitable for publication in PLOS ONE. Congratulations! Your manuscript is now with our production department. 

Kind regards, 

on behalf of

Dr. Alvaro Jose Mejia-Vergara 

Academic Editor

PLOS ONE